# The Quantitative Skeletal Muscle Ultrasonography in Elderly with Dynapenia but Not Sarcopenia Using Texture Analysis

**DOI:** 10.3390/diagnostics10060400

**Published:** 2020-06-12

**Authors:** Kuen-Cheh Yang, Yin-Yin Liao, Ke-Vin Chang, Kuo-Chin Huang, Der-Sheng Han

**Affiliations:** 1Department of Family Medicine, National Taiwan University Hospital, Bei-Hu Branch, Taipei 108, Taiwan; quintino.yang@gmail.com (K.-C.Y.); bretthuang@ntu.edu.tw (K.-C.H.); 2Community and Geriatric Medicine Research Center, National Taiwan University Hospital, Bei-Hu Branch, Taipei 108, Taiwan; kvchang011@gmail.com; 3Health Science and Wellness Center, National Taiwan University, Taipei 106, Taiwan; 4Department of Biomedical Engineering, Hungkuang University, Taichung 433, Taiwan; g9612536@gmail.com; 5Department of Physical Medicine and Rehabilitation, National Taiwan University Hospital, Bei-Hu Branch, Taipei 108, Taiwan; 6Department of Physical Medicine and Rehabilitation, National Taiwan University College of Medicine, Taipei 100, Taiwan; 7Department of Family Medicine, National Taiwan University College of Medicine, Taipei 100, Taiwan

**Keywords:** aged, hand strength, muscle strength, sarcopenia, ultrasonography

## Abstract

(1) Background: Dynapenia is defined as lower muscle strength alone. Only a few studies have investigated muscle quality in subjects with dynapenia. (2) Methods: The muscle quality, characterized by texture parameters of biceps brachii, triceps brachii, rectus femoris, and medial gastrocnemius muscles, were collected using ultrasonography. The risk of dynapenia was assessed by the multiple logistic regression model. (3) Results: There were a total of 36 participants (72.7 ± 5.8 yrs, 1:1 case-control matched). The values of texture parameters of autocorrelation (AUT) and sum variance (SVAR) in all four muscles were higher in the dynapenia group significantly (*p* < 0.05). AUT and SVAR had the significant associations for dynapenia in biceps (dds ratio[OR]:2.51, 95% CI = 1.25–5.07 for AUT; OR = 1.45, 95% CI:1.1–1.91 for SVAR), triceps (OR: 2.48, 95% CI = 1.60–5.3 for AUT; OR: 1.57, 95% CI = 1.08–2.28 for SVAR), and rectus femoris (OR: 1.58, 95% CI = 1.01–2.46 for AUT; OR: 1.2, 95% CI = 1.0–1.44 for SVAR). The areas under the receiver-operating curves of all texture parameters was between 0.84–0.94 after adjusting confounding factors. (4) Conclusions: The muscle quality in the dynapenia can be detected by the texture-feature quantitative ultrasound. Ultrasound measurement in the aging muscle might be promising, and further studies should validate its application in the context of dynapenia.

## 1. Introduction

The growth of an aging population has led to increased interest from scientific and medical societies on the research of aging. Skeletal muscle dysfunction, such as weakness, atrophy or poor coordination, is an important risk factor for quality of life and mortality in the elderly. Previous studies have shown that aged people have a loss of muscle mass and muscle strength at a rate of 0.64–0.98% and 2.5%–4% per year, respectively [1]. In recent years, there has been growing attention to the sarcopenia and consensuses on definition have been developed in Europe [2], Asia [3] and North America [4]. Most of the diagnostic criteria of sarcopenia include muscle strength, physical performance and muscle mass. However, Clark and Manini argued that the loss of muscle mass and strength need to be defined independently and proposed the term “dynapenia” to describe the age-related loss of muscle strength [5].

The mechanisms of dynapenia are attributable to neurological and muscular mechanisms. In terms of neural factors, dynapenia is related to aging-related cortical and spinal cord change with hypoexcitability. The reduction of the intrinsic force-generating capability and excitation–contraction coupling and increasing intramuscular fat with aging are the leading muscular causes of dynapenia. Accordingly, the muscle quality, instead of quantity, would be an important factor for older populations with dynapenia [6,7]. In recent years, several technologies used to evaluate muscle have become available, such as computed tomography (CT), magnetic resonance imaging (MRI), dual-energy X-ray absorptiometry (DXA), bioelectrical impedance analysis (BIA) and ultrasonography [8]. Among them, CT and MRI are expensive and mostly available due to research reasons, instead of general clinic use. DXA is less expensive but is not widely convenient for daily practice. BIA is inexpensive and less time-consuming, but its accuracy strongly relies on estimated equations and fluid status. Ultrasonography can be adhered to the specific training of operators and strict examination protocol. Accordingly, it could show a high degree of reliability among different raters and concordance to the results of MRIs. Furthermore, muscle ultrasound can also provide qualitative information about skeletal muscle, compared to the usual CT, MRI and DXA [9].

The muscle ultrasound assessment in the elderly includes the measurements of fascicle length, pennation angle, muscle thickness and echo intensity (EI) [9,10]. Most of this research was conducted in the populations of sarcopenia, which combines the low muscle mass and muscle weakness or poor physical performance. There are limited studies involving dynapenia, and the EI was calculated by the mean (first-order statistics) in previous studies. Our previous study [11] also showed that the participants of dynapenia, with normal skeletal muscle mass, had a decreased thickness of the rectus femoris and medial gastrocnemius muscles. However, the difference of muscle EI was not significant. Although there are significant differences of EI in previous studies of sarcopenia [12], they might not be sensitive enough to be used in the subjects of dynapenia with normal skeletal muscle mass. EI is rather subjective and depends on the skills and experience of the operator. State-of-the-art quantitative ultrasonography, which calculates the first and higher-order statistics according to the probability density function of grayscale, is widely available. The texture analysis of skeletal muscle ultrasonography has been shown to provide an objective and reliable method for differentiation between healthy and diseased muscles, such as Duchenne muscular dystrophy [13] and Pompe disease [14]. Quantitative texture features would represent the orientation and distribution of the echo patterns in muscular structures. However, it has not been used in the studies of sarcopenia or dynapenia. 

Therefore, this study aimed to assess the associations between the dynapenia and muscle quality using the texture analysis of ultrasonography in the elderly with normal skeletal muscle mass. We also tasked ourselves with finding which texture parameter could distinguish the dynapenia from the subjects with normal skeletal mass.

## 2. Materials and Methods

### 2.1. Participants and Grouping

During March 2016 to September 2016, an annual health examination for the elderly was conducted in a community hospital of Taipei, Taiwan. Exclusion criteria included walking with assistance, inability to follow instructions/fill the questionnaire, current malignancy with treatment course, hematological disorder, a recent admission history (3 months prior to recruitment) and dominant weight change (≥ 3% of original weight). In total, 140 elder people were enrolled. Individual information about current and past medical history, frequency of exercise, cigarette smoking, and betel nut chewing was obtained. The exercise was classified as regular (exercise ≥ 150 min per week) and not regular (no exercise, or exercise time < 150 min per week). Smoking status, alcohol consumption and betel nut chewing were defined as current and previous/never. All the participants provided written informed consent. The study protocol was approved by the Institutional Review Board of National Taiwan University Hospital on May 13, 2016 (IRB NO. 201601091RIND).

Among 140 participants, there were only 18 people in the dynapenia group (12.6%) with normal skeletal muscle mass. Accordingly, we randomly selected 18 subjects from the non-dynapenia group, after sex and age matching, as the reference group. 

### 2.2. Anthropometric Measurement

A standard electronic weight-and-height meter was used to measure weight and height with an accuracy of 100 g and 1mm, respectively. Body mass index (BMI) was calculated as the weight divided by height squared (kg/m^2^). Whole-body dual-energy X-ray absorptiometry (DXA, Stratos dR, DMS Group, France) was used to measure body composition. The skeletal muscle index (SMI) was determined by the total of four limb’s lean soft tissue (bone-free and fat-free mass, kilogram) divided by the square of the height (m^2^).

### 2.3. Strength and Physical Performance

All participants were asked to measure the muscle strength of their dominant hand. An isometric dynamometer was used when the participants seated with their forearm in a neutral position and elbow flexed at 90°. Three trials of muscle strength were measured by squeezing the handgrip device forcefully. The maximal value was recorded as the muscle strength.

The physical performance was measured by using gait speed of 5-meter walking test. Four metal strips marked the distance of the origin, 1 m, 6 m and 7 m on the ground. Each participant was instructed to walk as fast as possible in a comfortable manner. The time spent from 1m to 6m was recorded and gait speed was calculated accordingly (meter/second).

### 2.4. Ultrasound Measurement Protocol

All of the ultrasound images were obtained by using a multifrequency linear transducer (UP 200, BenQ Medical Technology Corp., Taipei, Taiwan). Four muscles (the biceps brachii, triceps brachii, rectus femoris, and medial gastrocnemius muscles) were examined with light touch and abundant jelly on the skin. The transducer has a central frequency of 7.5 MHz and can be automatically adjusted to 15 MHz when scanning the superficial target. We targeted the frame rate at 19 and 15 MHz for examination of the upper and lower extremities, respectively. The depth of the scanning widow was 3 cm for the upper limbs and 5 cm for the lower limbs. To resolve the dependence of the system settings in texture analysis, the instrument settings were standardized when imaging each subject, including identical gain, time gain compensation, and other relevant parameters. A constant level of gain in the grayscale was implemented for each ultrasound picture. All the scanning was carried out by a board-certificated physical therapist. All the muscles were measured in their short-axis for one time. We put the transducer midway between the elbow crease and the humeral head to measure the biceps muscle and at the midpoint of the posterior arm to measure the triceps brachii muscle. Likewise, we placed the transducer at the midpoint of the anterior thigh to examine the rectus femoris muscle and at the proximal one-third of the leg to examine the medial gastrocnemius muscle. Using the abovementioned protocol, the inter-rater and intra-rater reliabilities were 0.751 (95% CI: 0.472–0.894) and 0.835 (95% CI, 0.630–0.931), respectively, in a total of 20 muscles from five healthy adults according to our pilot test [11].

### 2.5. Texture Analysis of Ultrasound Image

The texture analysis of ultrasound images was based on a gray-level co-occurrence matrix (GLCM) of quantitative description of the distribution of co-occurring of gray-level values [15,16]. The GLCM provides a second-order method for generating texture features. Each entry (*i*, *j*) in GLCM counted the co-occurrence of pixels with gray level value *i* and *j* at given distance. In this study, the distance was set to 1, and the 0°, 45°, 90°, and 135° distinct orientations of neighboring pixels were selected. Number of gray levels (*G*) was denoted by 8 bit. Let *p*(*i*, *j*) be the (*i*, *j*) in a normalized GLCM. The mean for the rows and columns of the matrix are
(1)$$\mu_{x}=\overset{G}{\underset{i}{\sum }}\overset{G}{\underset{j}{\sum }}i\cdot p\left(i,j\right)\;$$
(2)$$\mu_{y}=\overset{G}{\underset{i}{\sum }}\overset{G}{\underset{j}{\sum }}j\cdot p\left(i,j\right)\;$$

Haralick proposed 14 statistical features extracted from GLCM to describe the texture of the images [17]. Subsequently, we used nine textural features of them from the region of interest (ROI) of the muscle image in our study. The examiner defined a rectangular ROI, which would fit within the muscle without extending beyond the borders of the muscle, ensuring no blood vessels. The following equations define these features [17,18,19,20,21].

(a)Autocorrelation (AUT) refers to repeating patterns of gray levels, measuring amount of regularity as well as the fineness/coarseness of texture.
(3)$$AUT=\overset{G}{\underset{i}{\sum }}\overset{G}{\underset{j}{\sum }}\left(i\cdot j\right)\cdot p\left(i,j\right)\;$$
(b)Contrast (CON) measures the local intensity contrast between a pixel and its neighbor over the whole image. Low CON values indicate the similar gray levels of each pixel pair.
(4)$$CON=\overset{G}{\underset{i}{\sum }}\overset{G}{\underset{j}{\sum }}{\left|i-j\right|}^{2}p\left(i,j\right)\;$$
(c)Cluster prominence (CPR) measures the grouping of pixels to characterize the clustering tendency of the pixels.
(5)$$CPR=\overset{G}{\underset{i}{\sum }}\overset{G}{\underset{j}{\sum }}{\left(i+j-\mu_{x}-\mu_{y}\right)}^{4}p\left(i,j\right)\;$$(d)Dissimilarity (DIS) measures the distance between pairs of pixels and is conceptually similar to the CON feature.
(6)$$DIS=\overset{G}{\underset{i}{\sum }}\overset{G}{\underset{j}{\sum }}\left|i-j\right|p\left(i,j\right)\;$$(e)Energy (ENE) measures the repetition of the pixel pairs to represent textural uniformity. High ENE values occur when the occurrence of repeated pixel pairs is high.
(7)$$ENE=\overset{G}{\underset{i,j}{\sum }}p{\left(i,j\right)}^{2}\;$$(f)Entropy (ENP) measures the randomness of gray level distribution to describe the degree of complication of textures of an image. The randomly distributed gray levels tend to have high ENP values.
(8)$$ENP=\overset{G}{\underset{i}{\sum }}\overset{G}{\underset{j}{\sum }}p\left(i,j\right)\;\log p\left(i,j\right)$$(g)Homogeneity (HOM) measures the local homogeneity of a pixel pair. The HOM value is high if the gray level differences in the pixel pairs are small.
(9)$$HOM=\overset{G}{\underset{i}{\sum }}\overset{G}{\underset{j}{\sum }}\frac{1}{1+{\left|i-j\right|}^{2}}p\left(i,j\right)\;$$(h)Maximum probability (MAXP) is retrieved from maximum value in the pixel pair. High MAXP values represent the high occurrence of the most predominant pixel pair.
(10)$$MAXP=\max.p\left(i,j\right)\;for\;all\left(i,j\right)\;$$(i)Sum variance (SVAR) measures the dispersion of the gray levels and is mathematically equal to the cluster tendency feature.
(11)$$SVAR=\overset{2G}{\underset{i=2}{\sum }}{\left(1-\mu \;\right)}^{2}p\left(i,j\right)$$

### 2.6. The Definition of Dynapenia Without Sarcopenia

Dynapenia was defined by the strength loss of handgrip with normal skeletal muscle mass. The cut-off value of low handgrip was 30kg for men and 20kg in women [2]. The normal skeletal muscle mass was defined as SMI > 7.40 kg/m^2^ for men and > 5.14 kg/m^2^ for women [22]. Only the subjects who had dynapenia and normal SMI were included in this study. 

### 2.7. Statistical Analysis

The continuous and categorical variables were shown by the mean (standard deviation, SD) and number (%), respectively. The differences between the dynapenia and non-dynapenia groups were examined by an independent t-test for continuous variables and Chi-square test for categorical variables. The association between dynapenia and the texture parameter of skeletal muscle were elucidated by the logistics regression model after adjusting for the age, BMI and SMI. The odds ratio (OR) with a 95% confidence interval (CI) of dynapenia was estimated for the texture parameters. Some parameters were not included in the logistics regression model when the 95% CI of the variable could not be converged. Receiver-operating curve (ROC) and the area under the ROC (AUC) with 95% of these parameters were both also estimated to show the diagnostic ability of dynapenia. Youden’s index was also used to identify the best cut-off-value to diagnose dynapenia. Accordingly, we also showed the sensitivity, specificity, accuracy, positive predictive value (PPV), negative predictive value (NPV), positive likelihood ratio, and negative likelihood ratio of texture parameters based on the best cut-off value.

The Pearson correlation coefficients (γ) were used to assess the association between texture parameters and anthropometric measurements. We assessed the variable associations with muscle strength measures of handgrip. A multiple linear regression model of handgrip was constructed with a priori explanatory variables of age, gender and SMI. A significant improvement of muscle texture parameters in the regression model was based on the change of R^2^ and examined by partial F test. A *p*-value <0.05 was defined as statistically significant. All analyses were conducted using SAS version 9.4 (SAS Inc., Cary, NC, USA).

## 3. Results

The gender, age, waist, BMI and health behaviors were not significant between the dynapenia and non-dynapenia groups (Table 1). The five-meter gait speed was significant slower in the dynapenia group compared to those in non-dynapenia group (1.11 ± 0.31 m/s vs. 1.3 ± 0.25 m/s, P = 0.056). 

Figure 1 depicts the ultrasound B-mode images of the biceps brachii, triceps brachii, rectus femoris, and medial gastrocnemius muscles in a normal subject and a patient with dynapenia, respectively. Compared with normal muscles, abnormal muscles affected by dynapenia are characterized by heterogeneous structures with whiter appearance. The muscle regions (delineated by the dashed white line in Figure 1) were extracted to calculate texture features.

Table 2 shows the measurement value of different texture parameters. The values of autocorrelation (AUT) and sum variance (SVAR) were all higher in the dynapenia group among the four muscle groups significantly (*p* < 0.001 in the biceps and triceps, *p* < 0.01 in medial gastrocnemius and *p* < 0.05 in rectus femoris). The estimates of contrast (CON) and dissimilarity and entropy (ENP) were significantly higher in biceps and triceps, but not in the medial gastrocnemius and rectus femoris. Contrarily, the homogeneity estimations of upper extremity muscles were significantly lower in the dynapenia group (*p* < 0.01 in biceps and *p* < 0.05 in triceps). The values of cluster prominence (CPR) and maximum probability (MAXP) were significantly different in the triceps and medial gastrocnemius (*p* < 0.05).

Table 3 shows the odds ratio (OR) (95% CI) of texture parameters (AUT, CPR and SVAR) for the risk of dynapenia. The other texture parameters were not shown here because the 95% CI of them could not be converged in the logistics regression model. AUT and CPR had the best performance to predict dynapenia. AUT had significant associations with dynapenia in biceps (OR = 2.51. 95% CI = 1.25–5.07), triceps (OR = 2.48, 95% CI = 1.60–5.3), and medial gastrocnemius (OR = 1.58, 95% CI = 1.01–2.46). Meanwhile, SVR also had significant associations with dynapenia in biceps (OR = 1.45, 95% CI = 1.10–1.91), triceps (OR = 1.57, 95% CI = 1.08–2.28), and medial gastrocnemius (OR = 1.20, 95% CI = 1.00–1.44). Dynapenia was associated with CPR in the triceps (OR = 1.01, 95% CI = 1–1.02) and rectus femoris (OR = 1.01, 95% CI = 1–1.01). The areas under the ROCs (AUCs) of all texture parameters were between 0.84–0.94 after adjusting for age, BMI and skeletal muscle index (SMI). The sensitivity, specificity, accuracy, positive predictive value (PPV), negative predictive value (NPV), positive likelihood ratio, and negative likelihood ratio of texture parameters based on the best cut-off-value (Youden’s index) were also shown in Appendix A
Table A1. We further checked the correlations between texture parameters and different muscles (Table 4). SMI and BMI were not correlated with any texture parameters. Similarly, increased handgrip was significantly correlated to AUT in all muscle groups (r = −0.46, −0.46, −0.49 and −0.47 for biceps, triceps, medial gastrocnemius and rectus femoris, respectively, *p* < 0.01). It was also significantly correlated to SVR in all muscle groups (r = −0.47, −0.44, −0.5 and −0.46 for biceps, triceps, medial gastrocnemius and rectus femoris, respectively, *p* < 0.01). Handgrip was correlated negatively to CPR only in triceps (r = −0.34, *p* < 0.05). The spent time of the five-meter walking was only positively correlated to AUT (r = 0.35, *p* < 0.05) and SVAR (r = 0.35, *p* < 0.05) in medial gastrocnemius.

Table 5 shows that the multiple regression model with age, gender and SMI as predictors of handgrip yielded adjusted R^2^ of 0.37. The addition of the AUT of biceps, triceps, or medial gastrocnemius significantly improved the model [ΔR^2^ = 0.135 (P = 0.004); 0.095 (P = 0.016) and 0.067 (P = 0.01), respectively]. The addition of the SVAR of biceps, triceps, or medial gastrocnemius significantly improved the model [ΔR^2^ = 0.145 (P = 0.003); 0.096 (P = 0.016) and 0.112 (P = 0.01), respectively]. However, the addition of the AUT or SVAR of rectus femoris did not improve the model significantly (P = 0.064 and P = 0.087, respectively). The addition of the CPR of any muscle did not yield an improvement for handgrip prediction. 

## 4. Discussion

This is the first study to apply the texture analysis of muscle in subjects with dynapenia. In our previous studies, although the decreased thicknesses of the rectus femoris and medial gastrocnemius muscles were significantly associated with dynapenia, the echo intensity of muscles was not associated with dynapenia [11]. We also demonstrated that the texture analysis of muscle could distinguish the differences of muscle quality between dynapenia and non-dynapenia in the subjects with normal skeletal muscle mass. Furthermore, the statistical model revealed the best variable of texture analysis for dynapenia might be autocorrelation (AUT) and sum variance (SVAR). 

Normal muscle tissue usually appears as a fine structure with hypoechoes and the fibroadipose septa or the sheath appears parallel to hyperechoes. In neuromuscular disorders, muscle tissue has heterogeneous structures and is more echogenic than usual, because it may be replaced by fibrosis and fat, leading to many transitions with different acoustic impedance, increasing the number of acoustic beam reflections [23]. However, the early stages of a neuromuscular disorder (such as dynapenia) could show only a little structural abnormality, resulting in only slightly increased echogenicity, which was barely detectable [23]. In this study, we demonstrated that the texture analysis of muscle could distinguish the differences of muscle between dynapenia and non-dynapenia in the subjects with normal skeletal muscle mass. Furthermore, the statistical model revealed the best variables of texture analysis for dynapenia might be AUT and SVAR. In contrast to normal muscles, abnormal muscles affected by dynapenia showed higher AUT values, representing a higher amount of coarseness of image texture; and higher SVAR values, representing greater asymmetry and a larger variation in gray levels.

In recent years, the development of high-resolution ultrasonography enables the easier evaluation of muscle, including lowering the cost, increasing availability at the clinic and improving the quickness of examination. When an aging society emerges, it also brings research interests on the aging process of muscle. Numerous epidemiological studies have reported sarcopenia as a risk of mortality and disability [24]. The European Working Group on Sarcopenia in Older People (EWGSOP) defines “sarcopenia” as the presence of both low muscle strength/physical function and low muscle mass [2]. Manini and Clark proposed that the loss of strength, the terminology of dynapenia, should be independent of sarcopenia due to the different entities of age-related loss of muscle mass and strength [5,6]. Recently, not only has the methodology to measure muscle mass come under debate [25], but lean muscle mass is no longer the most relevant feature to mortality compared with muscle strength [26]. Accordingly, the muscle mass and the muscle quality indicators were both important for the assessment of ageing-related muscle disorders. Regardless of the controversy between sarcopenia and dynapenia, a technique applied for the assessment of muscle mass and muscle quality at the same time would be promising.

The use of qualitative diagnostic ultrasound for muscle disorders has been reported in Duchenne muscular dystrophy [13], Pompe disease [14,27] and other inherited muscular disorders [28]. Recently, Ticinesi et al. reviewed the published studies on detecting muscle mass loss in older populations [9]. Many studies compared muscle ultrasound versus DXA, CT, MRI or BIA using muscle thickness, the cross-sectional area (CSA) of muscle in healthy volunteers or specific patients of heart, lung or liver diseases. Most of the studies supported the significant correlations (r = 0.52 to 0.99) between ultrasound parameters (thickness and CSA) and other image tools. The new structure parameters of fascicle length, pennation angle (e.g., the fiber of gastrocnemius medialis insertions into deep and supper aponeurosis) and derived physiological cross-sectional area (e.g., muscle volume divided by fascicle length) were also developed. Although fascicle length and pennation angle might decrease with aging [29], the differences between the elderly versus young people, or subjects with sarcopenia versus those without sarcopenia were inconsistent [10,30,31].

Nowadays, echo intensity (EI) is a state-of-the-art tool used to measure the muscle of elderly people. EI is calculated from the grayscale value (0–255) of targeted muscles. Several studies have demonstrated EI to be higher with aging and be related to many physical performances. Almost all studies measured the EI of lower-extremity muscles (quadriceps muscle or rectus femoris alone) and showed that EI was correlated negatively to muscle strength, physical or cardiovascular performance, such as 30-s sit-to-stand-up test, gait speed and oxygen uptake [32,33,34,35,36,37]. These findings were consistent with our previous study [11]: the EIs of biceps brachii, triceps brachii, and rectus femoris muscles were inversely correlated with handgrip strength (r = −0.253, −0.308 and −0.353, all *p* < 0.05). However, very few studies have discussed the comparisons of muscle quality between dynapenia and non-dynapenia. 

Yamada et al. conducted the comparison of EI of the rectus femoris muscle between low muscle function and normal muscle function [12]. Subjects with low muscle function not only had significant lower muscle thickness but also higher EI of the rectus femoris muscle. Contrarily, our previous studies showed that no association was identified between dynapenia and muscle EI [28]. The first possible reason is that the low muscle function in Yamada’s study was defined as low walking speed and/or low handgrip. The etiology of poor walking speed is a multifactorial and complex process, which is not only due to low muscle strength and quality [6]. Secondly, the SMI of normal muscle function in women was higher than those of low muscle function (6.2 ± 0.7 vs. 5.89 ± 0.78, *p* < 0.0001) in Yamada’s study. Higher EI partially came from the higher muscle mass. By contrast, the dynapenia was only defined by pure low handgrip alone, and the subjects of two groups had similar SMI in our study. This study found that the skeletal muscle texture characteristics of dynapenia are markedly different from non-dynapenia. However, traditional muscle EI failed to distinguish these changes [28]. This is the first study to demonstrate that using texture parameters from higher-order statistics of ultrasound images can delineate minor muscle structure changes. In this study, we sampled each muscle of biceps brachii, triceps brachii, rectus gastrocnemius and rectus femoris. Although dynapenia was dependent on the handgrip strength, the value of these textures between dynapenia and non-dynapenia was not only significantly different in the upper extremities, but also in the lower extremities, especially in terms of AUT and SVAR. These results implied that when the elderly had low handgrip strength, microstructure changes had already taken place in four extremities.

In this study, the texture parameters of AUT and SVAR in upper and lower extremities were correlated to handgrip (Table 3). Meanwhile, both of them had good diagnostic ability for dynapenia (Table 4, and Appendix A
Table A1). Furthermore, the structure changes also affected the walking performance. Therefore, the subjects with dynapenia had slower gait speed (m/s) than those without dynapenia (1.11 ± 0.31 m/s vs. 1.3 ± 0.25 m/s, P = 0.056 in Table 1). These findings are compatible to aging change in dynapenia, including the increasing intermuscular and intramuscular adipose tissue infiltration [15,23], and interstitial fibrous tissue [16]. Moreover, older adults with dynapenia with a compromised motor cortex or spinal cord and poor motor unit coupling leads to the impairment of all muscle groups. We speculate that the conclusion might be the same if dynapenia was defined by the muscle strength of knee extension instead of handgrip. Ismail et al. found that grip strength was associated with age and EI (adjusted R^2^ = 0.53) instead of SMI [38]. The partial F test for the addition of EI is significant (P = 0.017). Our study results also supported the theory that the muscle quality and microstructure account for the muscle strength (Table 5). The partial F tests for the addition of AUT or SVAR of any muscle were statistically significant (except in the rectus femoris, P = 0.064 for AUT and P = 0.087 for SVAR). The exception of the rectus femoris might be due to the small sample size in our study. Compared to previous studies, this is the first study supporting the use of texture parameters of skeletal muscle, which can elucidate the muscular mechanism or structure abnormality of muscle weakness (dynapenia). Since aging increases adipose tissue infiltration and interstitial fibrosis in muscle, an increase in muscle heterogeneity occurs. From the equation of the texture parameters, AUT refers to the repeating patterns of gray levels, the measuring amount of regularity and SVAR measuring the dispersion of the gray levels. A higher AUT signifies a greater amount of fineness/coarseness of texture, and a higher VAR indicates a larger difference between gray-level values and the corresponding global means. AUT and SVAR can reflect the aging effects on the heterogeneity of muscle quality. Current clinical evidence supports that sarcopenia may be secondary to the effects of dynapenia [6]. Furthermore, our previous study found that no association existed between EI and dynapenia. Both AUT and SVAR can reflect the degree of heterogeneity of muscle (second-order statistics) compared with EI showing the mean value of grey levels (first-order statistics). We can postulate that the texture parameters of AUT and SVAR represent the pre-status (dynapenia) with microstructure change of the muscle before EI change. In other words, the abnormal texture parameters symbolize the future application of ultrasound on the clinical evaluation or explore the possible mechanism of dynapenia in the elderly. Nevertheless, how the quantitative texture analysis is incorporated into the diagnosis of dynapenia needs further research and professional consensus.

There were some limitations in this study. First, we did not sample all the limb muscles and the database needs to be enriched to use the texture-feature parameters of different skeletal muscles. However, we performed the first study to demonstrate the associations between the muscle texture parameters of the biceps and triceps brachii and handgrip. Secondly, all the ultrasound images were performed by one examiner. Therefore, the inter- and intra-rater agreement of ultrasound assessment in different muscles should be studied in future studies. Third, since the number of subjects with dynapenia was few, we used the study design of case-control to overcome this limitation. Fourth, texture features were highly affected by the region of interest (ROI) size and system settings. The radio-frequency data of ultrasound backscattering will be analyzed and combined with texture analysis to avoid the effects of system and operator parameters. Moreover, instead of selecting specific ROIs, the more objective analysis using a segmentation of the whole muscle layer should be studied in the future. Nevertheless, the application of texture parameters in general elderly populations needs further study.

## 5. Conclusions

In conclusion, our study demonstrated that texture-feature quantitative imaging can be used to evaluate the muscle quality in the elderly with dynapenia. Although the current evidence cannot make conclusive recommendations regarding the use of muscle ultrasound in routine geriatric clinical assessments, the texture parameters may serve as a potentially useful diagnostic tool for cases of aging muscle. 

## Figures and Tables

**Figure 1 diagnostics-10-00400-f001:**
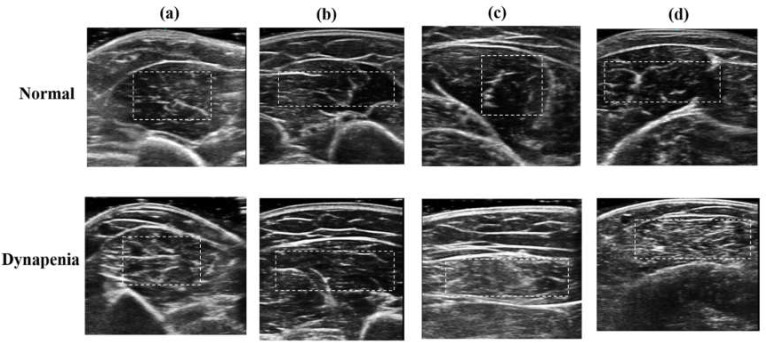
Different muscles of ultrasound B-mode images of a normal subject and a patient with dynapenia: (**a**) biceps brachii muscles, (**b**) triceps brachii muscles, (**c**) rectus femoris muscles, and (**d**) medial gastrocnemius muscles. The white dashed lines delineating the region of interest were manually traced by the physician.

**Table 1 diagnostics-10-00400-t001:** Basic demographics characteristics.

	Total	Non-Dynapenia	Dynapenia	*p*-Value
	Mean (SD)	Mean (SD)	Mean (SD)	
N (%)	36	18 (50%)	18 (50%)	
Demographics				
Men, n (%)	8 (22.2%)	4 (22.2%)	4 (22.2%)	1.000
Age (years)	72.69 (5.80)	70.88 (4.17)	74.50 (6.71)	0.060
Waist (cm)	80.53 (8.53)	78.51 (8.59)	82.55 (8.20)	0.158
BMI (kg/m^2^)	23.58 (3.37)	22.78 (3.50)	24.39 (7.31)	0.702
Physical performance				
Handgrip (kg)	24.14 (7.57)	29.28 (6.32)	19.00 (4.72)	<0.0001
Gait speed (m/s)	1.21 (0.30)	1.30 (0.25)	1.11 (0.31)	0.056
SMI (kg/m^2^)	6.19 (0.83)	6.15 (0.77)	6.24 (0.91)	0.751
Health behavior	N (%)	N (%)	N (%)	***p*-value**
Smoke (current)	1 (2.8%)	1 (5.6%)	0 (0.0%)	1.000
Alcohol (current)	9 (25%)	6 (33.0%)	3 (16.7%)	0.248
Betel nut (current)	2 (5.6%)	1 (5.6%)	1 (5.6%)	1.000
Exercise (regular)	17 (47.2%)	10 (55.6%)	7 (38.9%)	0.317

Notes: BMI, body mass index, weight (kg)/[height(m)]^2^; SMI, skeletal muscle index = four limbs lean muscle mass (kg)/[height(m)]^2^.

**Table 2 diagnostics-10-00400-t002:** Comparison of muscle texture parameters in dynapenia without sarcopenia.

	Upper Extremity		Lower Extremity	
	Biceps Brachii	Triceps Brachii	Medial Gastrocnemius	Rectus Femoris
Autocorrelation (AUT)	+ ***	+ ***	+ **	+ *
Contrast (CON)	+ **	+ **	+	+
Cluster Prominence (CPR)	+	+ *	+ *	+
Dissimilarity (DIS)	+ **	+ **	+	+
Energy (ENE)	−	− *	−	−
Entropy (ENP)	2020 *	+ **	+	+
Homogeneity (HOM)	− **	− *	−	+
Maximum probability (MAXP)	−	− *	− *	−
Sum variance (SVAR)	+ ***	+ ***	+ **	+ *

Notes: Positive (+): the value of measurement was higher in dynapenia; Negative (-): the value of measurement was higher in non-dynapnia; * *p* < 0.05; ** *p* < 0.01, *** *p* < 0.001.

**Table 3 diagnostics-10-00400-t003:** The diagnostic performance of muscle texture parameter for dynapenia.

	Biceps Brachii		Triceps Brachii		Medial Gastrocnemius		Rectus Femoris	
	OR (95%CI)	AUC (95%CI)	OR (95%CI)	AUC (95%CI)	OR (95%CI)	AUC (95%CI)	OR (95%CI)	AUC (95%CI)
Autocorrelation	2.51 (1.24−5.07) **	0.94 (0.86−1.00)	2.48 (1.16−5.30) *	0.90 (0.81−0.99)	1.58 (1.01−2.47) *	0.86 (0.73−0.99))	1.56 (0.99−2.46)	0.85 (0.73−0.98)
Cluster prominence	1.01 (1.00−1.02)	0.84 (0.71−0.97)	1.01 (1.00−1.02) *	0.91 (0.80−1.00)	0.99 (0.99−1.01)	0.78 (0.63−0.94)	1.01 (1.00−1.01) *	0.84 (0.71−0.97)
Sum variance	1.45 (1.10−1.91) **	0.94 (0.87−1.00)	1.57 (1.08−2.28) *	0.91 (0.82−1.00)	1.20 (1.00−1.44) *	0.85 (0.71−0.99)	1.18 (0.99−1.39)	0.84 (0.71−0.97)

Notes: * < 0.05; ** < 0.01; *** < 0.001; OR: odds ratio; CI: confidence interval, AUC: area under the receiver operating curve; Model adjusted for age, BMI, exercise and SMI.

**Table 4 diagnostics-10-00400-t004:** Correlation coefficients (γ) between muscle texture parameters and anthropometric variables.

Muscle	Parameter	Age	BMI	SMI	Grip	Gait
Biceps brachii	AUT	0.18	0.02	−0.07	−0.46 **	0.16
	CPR	−0.09	−0.25	−0.18	−0.11	−0.33
	SVAR	0.17	0.06	−0.05	−0.47 **	0.17
Triceps brachii	AUT	0.20	−0.02	−0.17	−0.46 **	0.08
	CPR	−0.06	0.13	−0.04	−0.34 *	0.07
	SVAR	0.17	−0.001	−0.14	−0.44 **	0.08
Medial gastrocnemius	AUT	0.18	0.18	0.02	−0.49 **	0.35 *
	CPR	−0.54 ***	−0.09	−0.10	0.12	−0.24
	SVAR	0.18	0.19	0.01	−0.50 **	0.35 *
Rectus femoris	AUT	−0.05	0.09	−0.09	−0.47 **	−0.09
	CPR	−0.37	0.07	−0.07	−0.16	−0.30
	SVAR	−0.02	0.09	−0.09	−0.46 **	−0.09

Notes: AUT: Autocorrelation; CPR: Cluster prominence; SVAR: Sum variance; BMI: body mass index (kg/m^2^); SMI, skeletal muscle index (kg/m^2^); Gait: walk speed of 5-meter (m/s); Grip: handgrip, kg; *< 0.05; **< 0.01; ***< 0.001.

**Table 5 diagnostics-10-00400-t005:** The linear regression model to predict the grip strength.

Parameter		Adjusted R^2^		
		Model 1	Model 2	*p*-Value *
**Autocorrelation** **(AUT)**	Biceps brachii	0.371	0.506	0.004
	Triceps brachii		0.466	0.016
	Medial gastrocnemius		0.438	0.010
	Rectus femoris		0.422	0.064
Cluster prominence (CPR)	Biceps brachii	0.371	0.371	0.330
	Triceps brachii		0.425	0.058
	Medial gastrocnemius		0.353	0.749
	Rectus femoris		0.374	0.299
Sum variance (SVAR)	Biceps brachii	0.371	0.516	0.003
	Triceps brachii		0.467	0.016
	Medial gastrocnemius		0.483	0.010
	Rectus femoris		0.412	0.087

Notes: Model1: Adjust for gender, age, SMI.; Model 2: Model1 add echo parameter.; * *p*-value of partial F-test for the addition of echo texture variable.

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
