# Peer review of "The Quantitative Skeletal Muscle Ultrasonography in Elderly with Dynapenia but Not Sarcopenia Using Texture Analysis"

_diagnostics, 2020, doi:10.3390/diagnostics10060400_

Round 1
Reviewer 1 Report
Summary of study finding
The authors investigated the relation between the dynapenia and muscle quality using the texture analysis of ultrasonography in case- control study. They suggested that it may be useful to assess the value of texture parameters of autocorrelation and sum variance of muscles to detect the dynapenia. I think that the result of this study is properly, containing new information, however, I think the authors need some work and revise the manuscript.
Major comments
Title
Accurately reflects content of the manuscript.
Abstract
It can be understood without the content of the manuscript.
Introduction
Adequate and relevant.
Results
4 page:” Table 3 showed the OR (95%CI) of texture parameters (AUR,”
I would reword from AUR to AUT.
Discussion
8 page: line 66-67. The authors described the Yamada’s study (Reference No27). However, the article, that is Reference No27, is not Yamada’s study. Please check it. (It may be “Differential characteristics of skeletal muscle in community-dwelling older adults. Yamada M, 2017” ?)
9 page: line 113. “sarcopenia was few” I would reword from sarcopenia to dynapenia.
This study demonstrated the usefulness of the texture parameters of ultrasonography to diagnosis the dynapenia. What does the value of each texture parameter reflect the biological muscle reaction? (the degree of increasing intermuscular and intramuscular adipose tissue infiltration or interstitial fibrous tissue ?) Please clarify the meaning of the value of the texture parameters.
Materials and Methods
4.1Participants and grouping
10 page: line 123-124. The authors investigated the frequency of exercise, cigarette smoking, alcohol, betel nuts chewing. Please clarify the definition of the frequency of exercise, cigarette smoking, alcohol, betel nuts chewing.
10 page: line 127. 18 subjects of dynapenia were selected in this study. However, the criteria of dynapenia was unclear in this study. The authors should described the diagnostic criteria of dynapenia of this study.
4.4 Ultrasound measurement protocol
How many times dose the examiner scan the each muscle by ultrasonography?
Author Response
Reviewer 1
The authors investigated the relation between the dynapenia and muscle quality using the texture analysis of ultrasonography in case-control study. They suggested that it may be useful to assess the value of texture parameters of autocorrelation and sum variance of muscles to detect the dynapenia. I think that the result of this study is properly, containing new information, however, I think the authors need some work and revise the manuscript.
Major comments
Point 1: Title: Accurately reflects content of the manuscript.
Response 1: Thank you for your comment.
Point 2: Abstract: It can be understood without the content of the manuscript.
Response 2: Thank you for your comment.
Point 3: Introduction: Adequate and relevant.
Response 3: Thank you for your comment
Point 4: Results: 4 page:” Table 3 showed the OR (95%CI) of texture parameters (AUR would reword from AUR to AUT.
Response 4: We sincerely apologized for this typo and have corrected this word.
Point 5: Discussion
8 page: line 66-67. The authors described the Yamada’s study (Reference No27). However, the article, that is Reference No27, is not Yamada’s study. Please check it. (It may be “Differential characteristics of skeletal muscle in community-dwelling older adults. Yamada M, 2017” ?)
Response 5: Yes. The correct reference should be “Yamada M, Kimura Y, Ishiyama D, et al. Differential Characteristics of Skeletal Muscle in Community-Dwelling Older Adults. J Am Med Dir Assoc. 2017;18(9):807.e9-807.e16.”
Point 6: 9 page: line 113. “sarcopenia was few” I would reword from sarcopenia to dynapenia.
Response 6: We apologized for this typo and have corrected this word from “sarcopenia” to “dynapenia”. (#Line 354)
Point 7: This study demonstrated the usefulness of the texture parameters of ultrasonography to diagnosis the dynapenia. What does the value of each texture parameter reflect the biological muscle reaction? (the degree of increasing intermuscular and intramuscular adipose tissue infiltration or interstitial fibrous tissue ?) Please clarify the meaning of the value of the texture parameters.
Response 7: We appreciated for this comment. In original manuscript, we have mentioned:
- Line 305-306: These findings were compatible to aging change in dynapenia, including intramuscular adipose tissue infiltration[15,30], and interstitial fibrous tissue[31].
In order to strengthen the value of this new finding, we amended original sentences to discuss the possible biological meaning of the texture parameters of Autocorrelation (AUT) and Sum variance (SVAR) as follow: (#Line 316-327)
Since aging increases adipose tissue infiltration and interstitial fibrosis in muscle, the increase in muscle heterogeneity occurs. From the equation of the texture parameters, AUT refers to repeating patterns of gray levels, measuring amount of regularity and SVAR measuring the dispersion of the gray levels. A higher AUT signifies a greater amount of the fineness/coarseness of texture, and a higher VAR indicates a larger difference between gray-level values and the corresponding global means. AUT and SVAR can reflect the aging effects on the heterogeneity of muscle quality. Current clinical evidences support that sarcopenia may be secondary to the effects of dynapenia[6]. Furthermore, our previous study found that no association existed between EI and dynapenia. Both of AUT and SVAR can reflect degree of heterogeneity of muscle (second-order statistics) compared with EI showing the mean value of grey levels (first-order statistics).We can postulate that the texture parameters of AUT and SVAR represent the pre-status (dynapenia) with microstructure change of the muscle before EI change.
Materials and Methods
4.1Participants and grouping
Point 8: 10 page: line 123-124. The authors investigated the frequency of exercise, cigarette smoking, alcohol, betel nuts chewing. Please clarify the definition of the frequency of exercise, cigarette smoking, alcohol, betel nuts chewing.
Response 8: We added the descriptions as follows: (#Line 368-370)
The exercise was classified as regular (exercise ≥ 150minutes per week) and not regular (no exercise, or exercise time < 150minutes per week). Smoking status, alcohol consumption and betel nuts chewing were defined current and previous/never.
Point 9: 10 page: line 127. 18 subjects of dynapenia were selected in this study. However, the criteria of dynapenia was unclear in this study. The authors should described the diagnostic criteria of dynapenia of this study.
Response 9: We did describe the definition of dynapenia at 4.6 The definition of dynapenia without sarcopenia (#Line 449-453).
Dynapenia was defined by strength loss of handgrip with normal skeletal muscle mass. The cut-off value of low handgrip was 30kg for men and 20kg in women[2]. The normal skeletal muscle mass was defined as SMI > 7.40 kg/m2 for men and > 5.14 kg/m2 for women[38]. Only the subjects who had dynapenia and normal SMI were included in this study.
Point 10: 4.4 Ultrasound measurement protocol: How many times dose the examiner scan the each muscle by ultrasonography?
Response 10: Because we have standardized protocol for the measurement for each muscle. Therefore, we scanned the target muscles only for one time. We amended the sentence as follows: (#Line 406)
All the muscles were measured in their short-axis for one time.
Reviewer 2 Report
General comment;
This submission should have line numbers to facilitate reviewers job.
Introduction;
Many sentences in the introduction are missing citing references.
Methods, results and discussion;
The authors have shown the diagnostic performance of muscle texture parameter for dynapenia. it might be useful if the authors could show more parameters related to AUC analysis such as Optimal cutoff point, Youden’s index (J), Accuracy (%), Sensitivity (%), Specificity (%), Positive predictive value (%), Negative predictive value (%), Positive likelihood ratios, Negative likelihood ratios.
Reviewer 3 Report
This study aimed to assess the associations between the dynapenia and muscle quality using the texture analysis of ultrasonography in the elderly with normal skeletal muscle mass and the dynapenia. Specifically, the analysis was based on a gray level co-occurrence matrix and Haralick’s texture features. Although the authors demonstrated that the dynapenia could be quantitatively identified by the texture analysis, some limitations (the number of patients, inter/intra-operator reliability, various region of interest) have been found.
- Although one examiner carried out the clinical test, the imaging parameters of an ultrasound system may vary. Therefore, the parameters for the texture analysis should be dependent on the imaging parameters. How to resolve the potential issue in texture analysis?
- What is the quantitative golden standard for the diagnosis of dynapenia? It would be great to discuss comparisons between the golden standard and your method.
- Why do you utilize a linear regression method for the classification? Any specific reasons? Authors may want to include the performance of other machine learning or statistical methods for the diagnosis.
- The outcomes from the analysis would be affected by specifying the various regions of interest in different images. For your future work, I would suggest more objective analysis via segmentation of the whole muscle layer (manually or automatically) instead of specifying the limited regions of interest.
- In the 1st paragraph of the result section, the p-value of gait speed is as smaller than 0.0001. However, it is 0.028 in Table 1. Please confirm it.
- In section 4.4, the frame rate and imaging frequency are not clear. What are the frame rate and imaging frequency? Please confirm. Please add the unit.
- In section 4.5, Equation(11), ‘sum variance’ is described but ‘sum of the square’ is described in [30]. Please confirm what feature had been used and correct it.
Author Response
Reviewer 3
This study aimed to assess the associations between the dynapenia and muscle quality using the texture analysis of ultrasonography in the elderly with normal skeletal muscle mass and the dynapenia. Specifically, the analysis was based on a gray level co-occurrence matrix and Haralick’s texture features. Although the authors demonstrated that the dynapenia could be quantitatively identified by the texture analysis, some limitations (the number of patients, inter/intra-operator reliability, various region of interest) have been found.
Point 1: Although one examiner carried out the clinical test, the imaging parameters of an ultrasound system may vary. Therefore, the parameters for the texture analysis should be dependent on the imaging parameters. How to resolve the potential issue in texture analysis?
Response 1: Thank you for this comment. We have standardized ultrasound protocol, including the process of taking the image of muscles and standardized system setting in this study. We also have good inter- and intra-rater reliabilities in our pre-test to compensate this shortcoming. However, the texture analysis still has some limitations compared to the original radiofrequency data. Therefore, we added some sentences in the section 4.4 Ultrasound Measurement Protocol and the limitation in discussion.
4.4 Ultrasound Measurement Protocol
#Line 402-404
To resolve the dependence of the system settings in texture analysis, the instrument settings were standardized when imaging each subject, including identical gain, time gain compensation, and other relevant parameters.
#Line 411-413
Using the abovementioned protocol, the inter-rater and intra-rater reliabilities were 0.751 (95% CI: 0.472-0.894) and 0.835 (95% CI, 0.630-0.931), respectively, in a total of 20 muscles from five healthy adults according to our pilot test [11].
Limitation in Discussion: (#Line 355-358)
The radio-frequency data of ultrasound backscattering will be analyzed and combined with texture analysis to avoid the effects of system and operator parameters. Moreover, instead of selecting specific ROIs, the more objective analysis using segmentation of the whole muscle layer should be studied in the future.
Point 2: What is the quantitative golden standard for the diagnosis of dynapenia? It would be great to discuss comparisons between the golden standard and your method.
Response 2: Dynapenia is a clinical diagnosis, which is defined by low strength of muscle mass with normal skeletal mass. Currently, the image study is not included into the diagnosis criteria. Our study results demonstrated that the texture parameters are still able to reveal microstructure changes in the muscle of subjects of dynapenia with normal skeletal mass. As a result, we extended the original discussion:
#Line 328-331
In other words, the abnormal texture parameters symbolized the future application of ultrasound on the clinical evaluation or explore the possible mechanism of dynapenia in the elderly. Nevertheless, how the quantitative texture analysis is incorporated into the diagnosis of dynapenia needs the further research and professional consensus.
Point 3: Why do you utilize a linear regression method for the classification? Any specific reasons? Authors may want to include the performance of other machine learning or statistical methods for the diagnosis.
Response 3: In the linear regression model, dependent variable is grip strength (table 5). We intended to show the explanatory power of the texture parameters (independent variables) of different muscle to predict grip strength. Therefore, the linear regression method was not utilized for the classification (diagnosis) of dynapenia. Furthermore, in the case-control trial, the number of the subject is too few to use machine learning.
Point 4: The outcomes from the analysis would be affected by specifying the various regions of interest in different images. For your future work, I would suggest more objective analysis via segmentation of the whole muscle layer (manually or automatically) instead of specifying the limited regions of interest.
Response 4: We appreciate your suggestions. We added the sentence in the limitation.
#Line 357-359
Moreover, instead of selecting specific ROIs, the more objective analysis using segmentation of the whole muscle layer should be studied in the future.
Point 5: In the 1st paragraph of the result section, the p-value of gait speed is as smaller than 0.0001. However, it is 0.028 in Table 1. Please confirm it.
Response 5: Thank you for pointing this error. We amended the results in the text and Table 1 as follows: (#Line 104-105)
The 5-meter gait speed was significant slower in the dynapenia group compared to those in non-dynapenia group (1.11 ± 0.31 m/s vs. 1.3 ± 0.25 m/s, P=0.056).
Point 6: In section 4.4, the frame rate and imaging frequency are not clear. What are the frame rate and imaging frequency? Please confirm. Please add the unit.
Response 6: In our original manuscript, we have described the frame rate and imaging frequency of the transducer as follows (Line# 396-401):
The transducer has a central frequency of 7.5 MHz and can be automatically adjusted to 15 MHz when scanning the superficial target. We targeted the frame rate at 19 and 15 MHz for examination of the upper and lower extremities, respectively.
Point 7: In section 4.5, Equation(11), ‘sum variance’ is described but ‘sum of the square’ is described in [30]. Please confirm what feature had been used and correct it.
Response 7: Both ‘sum of the square’ and ‘sum variance’ were described in the appendix in the reference [30]. We have confirmed that ‘sum variance’ was used in this study.
Round 2
Reviewer 1 Report
Thank you for your revisions of the article. I think it reads better than previously and the changes better convey the intended meaning. I feel qualified to publish in this journal.
Reviewer 2 Report
the authors covered all the reviewer comments.
Reviewer 3 Report
Authors well-addresed the raised concerns. I don't have any further comments on this manuscript.